# Properties of Organosilicon Elastomers Modified with Multilayer Carbon Nanotubes and Metallic (Cu or Ni) Microparticles

**DOI:** 10.3390/polym16060774

**Published:** 2024-03-11

**Authors:** Alexander V. Shchegolkov, Aleksei V. Shchegolkov, Natalia V. Zemtsova, Alexandre A. Vetcher, Yaroslav M. Stanishevskiy

**Affiliations:** 1Institute of Power Engineering, Instrumentation and Radioelectronics, Tambov State Technical University, 392000 Tambov, Russia; natasha_paramonova_68@mail.ru; 2Center for Project Activities, Moscow Polytechnic University, Bolshaya Semenovskaya St., 38, 107023 Moscow, Russia; alexxx5000@mail.ru; 3Institute of Biochemical Technology and Nanotechnology (IBTN), Peoples’ Friendship University, Russia n.a. P. Lumumba (RUDN), 6 Miklukho-Maklaya St., 117198 Moscow, Russia; stanishevskiy-yam@rudn.ru

**Keywords:** multi-walled carbon nanotubes, catalyst, organosilicon compound, modification, degradation, heat output, thermal field

## Abstract

The structural and electro-thermophysical characteristics of organosilicon elastomers modified with multilayer carbon nanotubes (MWCNTs) synthesized on Co-Mo/Al_2_O_3_-MgO and metallic (Cu or Ni) microparticles have been studied. The structures were analyzed with scanning electron microscopy (SEM), transmission electron microscopy (TEM), Raman spectroscopy, and energy-dispersive X-ray spectroscopy (EDX). The main focus of this study was the influence of metallic dispersed fillers on the resistance of a modified elastomer with Cu and Ni to the degradation of electrophysical parameters under the action of applied electrical voltage. The distribution of the temperature field on the surface of a modified polymer composite with metallic micro-dimensional structures has been recorded. The collected data demonstrate the possibility of controlling the degradation caused by electrical voltage. It has been found that repeated on/off turns of the elastomer with an MWCNTs on 50 and 100 cycles leads to a deterioration in the properties of the conductive elastomer from the available power of 1.1 kW/m^2^ (−40 °C) and, as a consequence, a decrease in the power to 0.3 kW/m^2^ (−40 °C) after 100 on/off cycles. At the same time, the Ni additive allows increasing the power by 1.4 kW/m^2^ (−40 °C) and reducing the intensity of the degradation of the conductive structures (after 100 on/off cycles up to 1.2 kW/m^2^ (−40 °C). When Ni is replaced by Cu, the power of the modified composite in the heating mode increases to 1.6 kW/m^2^ (−40 °C) and, at the same time, the degradation of the conductive structures in the composite decreases in the mode of cyclic offensives (50 and 100 cycles) (1.5 kW/m^2^ (−40 °C)). It was found that the best result in terms of heat removal is typical for an elastomer sample with an MWCNTs and Cu (temperature reaches 93.9 °C), which indicates an intensification of the heat removal from the most overheated places of the composite structure. At the same time, the maximum temperature for the Ni additives reaches 86.7 °C. A sample without the addition of a micro-sized metal is characterized by the local unevenness of the temperature field distribution, which causes undesirable internal overheating and destruction of the current-conducting structures based on the MWCNTs. The maximum temperature at the same time reaches a value of 49.8 °C. The conducted studies of the distribution of the micro-sizes of Ni and Cu show that Cu, due to its larger particles, improves internal heat exchange and intensifies heat release to the surface of the heater sample, which improves the temperature regime of the MWCNTs and, accordingly, increases resistance to electrophysical degradation.

## 1. Introduction

Polymer composites are widely used in various types of industrial applications [1]. This is due to their advantages compared to classical materials: metals and ceramics. Polymers possess flexibility, low density, ease of processing, and corrosion resistance [2] and can be used in conditions of increased vibration [3].

Electrically conductive polymer matrices with dispersed conductive fillers are in high demand [4,5]. Electrically conductive poly-dimensional matrices are characterized by a large range of functional properties [6] and, in particular, a good ability to absorb an electromagnetic field [7]. It should be noted that smart polymers are polymer composites also derived from conductive fillers [8,9]. Smart polymers can be used for health monitoring, deformation detection [8], as well as multi-faceted sensor applications [9]. Therefore, conductive polymer composites are used in many technological applications, such as weight sensors [10], anticorrosive films [11], electrochromic devices [12], and electric heaters with the effect of temperature self-regulation [13].

Carbon nanotubes (CNTs) [14], carbon black [15], graphite [16], graphene [17], graphene oxide [18], and dispersed metal nano- and microparticles [19,20] can be used as conductive fillers. Variations in conductive fillers due to changes in the concentration and type of filler allow for regulation of their electrical properties due to the formation of conductive grids of dispersed filler. Carbon and metal fillers can be combined in a polymer composite [21]. Polymer composites, which can basically contain combined conductive fillers, have technological significance and their application opens up new prospects for the development of smart materials [22,23,24,25,26].

The introduction of a dispersed metal into the polymer structure makes it possible to significantly improve several electro- and thermophysical properties of polymers [22]. The authors disperse AuNPs and CNT together in a polystyrene matrix. The conductivity of the composite is only 10^−4^ S/cm, with a mass ratio of AuNP to polystyrene of 1:2. Typically, the intrinsic conductivity of CNTs ranges from 10^4^ S/cm to 10^6^ S/cm. Another effective method for improving the conductivity of the CNT network is to combine particles of metals or alloys with high conductivity (such as Au and Cu (σ Au = 4.16 × 10^7^ S/cm and σ Cu = 5.88 × 10^7^ S/cm)) on the CNT surface [23,24]. It should be noted that Au is much more expensive, which hinders its employment at an industrial scale. Cu is much cheaper, but it must be pronged to be oxidized. In this regard, it is necessary to consider an AuCu alloy in which there will be Au in such an amount as to reduce the oxidation of Cu. In the synthesis of Ni nanoparticles on carbon tubes, the particle size can be well controlled [25].

The study of electrical conductivity shows that the films that are metal/CNT/PDMS are more conductive than CNT/PDMS films [26]. After experimental verification, a simulation model based on the theory of percolation networks and Monte Carlo technology is used. The calculation results also show that the conductivity of flexible composites can reach more than 100 S/cm in Ag/CNT/PDMS, CuAg/CNT/PDMS, Cu/CNT/PDMS, AuCu/CNT/PDMS or Au/CNT/PDMS films. Since the CNT load is 8% by weight, the metal particle coating coefficient is 100%, and the intrinsic conductivity of the CNTs reaches 10.6 S/cm.

One form of application of metal-dispersed additives in polymers is the use of microspheres in the form of polymer beads coated with Au or Ag. However, to ensure a strong adhesion between the polymer microspheres and noble metals, an Ni layer with a high non-existent capability is required, which leads to complex stages of pretreatment of the surface of the polymer particles [27]. Dispersed copper and nanowires can become an attractive alternative to Ag as conductive (nano)materials [28].

The choice of polymer matrix is also an important aspect in the production and application of functional composites [29]. Elastomers with flexibility and extensibility are suitable as a matrix for intelligent textile applications [30]. The best characteristics for elastomers are characteristic of matrices based on organosilicon compounds [31,32] and various types of polyurethane [33,34,35,36].

The addition of CNT to the polymer makes it possible to reduce the thermal degradation of the polymer matrix, which can be explained by the improvement in the distribution of heat fluxes in the volume of the polymer matrix [37]. In [38], the connection of morphology with local overheating is demonstrated, as well as CNT from the point of view of the stability of composites concerning thermal decomposition. The mechanism of CNT destruction under the action of a high current load causes local thinning [39]. When the inner wall itself is destroyed, a significant drop in current is observed, which proves that each wall is conductive and therefore affects the overall electrical conductivity of the carbon nanotube. At the same time, the degradation of some CNTs occurs not only in the center but also on the marginal side of the CNT [40]. It should be taken into account that the degradation of CNTs is associated with oxidative processes during the flow of electric current, whereas vacuum CNTs withstand higher current densities and reach maximum current-carrying capacity [41,42].

Organosilicon compounds are more flexible and deformable than other polymers used to create functional composites. Since the conducting behavior depends on the nature of the polymer and the interaction of the matrix with the filler, it is important to investigate the conducting behavior of composites with additives of micro-sized metals and carbon nanostructures (mainly CNTs).

The current study is focused on the management of the electrophysical properties of modified composites depending on the addition of metallic microparticles:Elastomers’ manufacturing with MWCNTs additives and micro-sized Ni and Cu, which have different sizes, as well as electrical conductivity.Analysis of the structural and morphological properties of MWCNTs and elastomers with MWCNT additives and Ni and Cu micro-particles.Studies of the influence of temperature conditions on the electrophysical processes in elastomers with additive MWCNTs and Ni and Cu micro-particles.Comparison of the distribution of thermal fields on the surface of modified composites with Ni and Cu micro-particles.

## 2. Materials and Methods

CVD technology was used for the MWCNTs synthesis (briefly: propane–butane mixture as a carbon source, t = 650 °C, 40 min). A multicomponent oxide system was used as a dispersed catalyst: Co-Mo/Al_2_O_3_-MgO with a scaly structure (Table 1).

The elastomer Silagerm 8030 (Element 14, LLC, Moscow, Russia), an organosilicon compound with polar Si–O bonds, was used as the basis of the polymer matrix. Samples of heaters were obtained by the method presented in [43,44]. The micro-sized powders (Ni or Cu) were from Plasmotherm LLC (Moscow, Russia).

The mass concentration of the MWCNTs was 3 wt.% and the microdispersed metals 5 wt.% for the Ni and Cu microparticles. The influence of various concentrations of MWCNTs and microdispersed metals was estimated based on data on the specific surface and volumetric resistance, taking into account the power of heat release by the action of electric voltage.

An ultrasonic dispergator UP400 St (Hielscher Ultrasonics GmbH, Berlin, Germany) was used for the distribution of MWCNTs and metals in the Silagerm 8030 organosilicon compound in a liquid state and the reduction of the number of agglomerates and aggregates.

The main stages of the modified composite’s manufacturing were:-mixing of components of organosilicon compounds A and B (component A/component B = 1/1);-calculation of the mass concentration of the MWCNTs in relation to the mass of the silicon–organic compound (A + B) in accordance with Equation (1):
(1)$$KK\;-\;MWCNTs\;=\;\frac{\left(MKK\;\times \;CMWCNTs\right)}{\left(100wt.\%\;-\;CMWCNTs\right)}$$
where KK−MWCNTs is the mass of the MWCNTs and KK, kg; MKK is the mass of the organosilicon compound (KK) (A + B), kg; and C MWCNTs is the mass concentration of MNT, wt.%;-calculation of the mass concentration of the metal additives (Ni or Cu) in relation to the mass of the organosilicon compound (A + B) (2):$KK\;-\;MWCNTs=\frac{\left(MKK\;\times \;Me\;(Ni\;or\;Cu)\right)}{\left(100\%\;by\;weight\;-\;Me\;(Ni\;or\;Cu)\right)}$ ((Figure 1a));-mechanical mixing of the MWCNTs and metal additives (Ni or Cu);-introduction of the calculated mass concentration of MWCNTs and metal additives (Ni or Cu) into a liquid organosilicon compound (A + B) (Figure 1b);-formation of samples of heaters with feeding electrodes (Figure 1c);-polymerization of an organosilicon compound with MWCNTs and metallic additives (Ni or Cu) at a temperature of (85 ± 5) °C in a vacuum drying unit for 5 hrs. In this case, a working prototype of the heating elements was produced by mechanically applying a polymer matrix to the surface of a foil electrode. Metal limiters were used to control the thickness of the active layer (elastomer modified by an MNT), and the resulting prototype was covered with glass for an even distribution of the polymer matrix and for better contact between the foil electrodes and the active layer (Figure 1c). Therefore, wires were soldered to the resulting heaters, and the open surface of the aluminum electrode was insulated in accordance with Figure 1d. The scheme for obtaining the materials and conducting the experiments is shown in Figure 1e. A programmable ATN 1351 power supply unit (Elix LLC, Moscow, Russia) with a control range from 0 to 300 V was used for power regulation.

### 2.1. Studies of the Temperature Field on the Surface of Samples of Modified Elastomers

To study the temperature field, a Testo-875-1 thermal imager with a 32 × 23° optical lens (SE & Co. KGaA, Testo, Lenzkirch, Germany) was used at a distance of 10 cm from the samples of nanomodified elastomers in a darkened room without exposure to sunlight. The temperature of the nanomodified elastomers was measured with a two-channel thermometer “Testo 992” (SE & Co. KGaA, Testo, Lenzkirch, Germany), while the surface temperature was determined, and based on the data obtained, a comparison was performed with the temperature recorded by the thermal imager, after which the emissivity selected was used for further measurements. The obtained thermal imaging images of the composite samples with MWCNTs were processed using the IRSoft v 5.0 SP1 program.

### 2.2. Structural Studies of MWCNTs and Elastomer’s Matrix

Structural studies were carried out with TEM and SEM. A small number of samples were made via contact with microscopic meshes with an adhesive composition. The studies were carried out from different places on the samples and on several samples in order to obtain better statistics about the samples under study. The TEM and SEM studies were carried out using a Hitachi H-800 (Hitachi Ltd., Tokyo, Japan) with an accelerating voltage up to 200 keV.

To study the morphology of the elastomer samples, the SEM method was used, which was implemented in the following device: Thermo Fisher Phenom XL G2 Desktop SEM. The EDX energy dispersion spectroscopy module was built-in. Software—Thermo Fisher (Waltham, MA, USA).

A Nikon Eclipse LV 150 optical binocular microscope (Nikon Solutions Co., Ltd., Nagoya, Japan) was also used to estimate the distribution of dispersed particles. A Fourier spectrophotometer FT 801 (Spectral range 21–1.8 µm) (LLC NPF Simeks, Novosibirsk, Russia) was used for the registration in the near- and mid-IR ranges of the spectra of the original and modified with MWCNT elastomer matrices. To estimate the particle sizes, the NICOMP 380 ZLS Zeta Potential/Particle Sizer laser particle analyzer (PSS.Nicomp, Santa Barbara, CA, USA) was used, the research method for which is based on the method of dynamic light scattering.

### 2.3. Raman Spectroscopy

A spectrometer based on a confocal microscope (Spectra, NT-MDT SI, Moscow, Russia) was used to measure the Raman spectra. It was used in two modes, providing signal collection from the same volume. A 100 × lens with NA = 0.7, a semiconductor laser (λ = 532 nm, excitation power of about 50 MW) and a point hole with a diameter of 100 microns provided a lateral resolution of about 400 nm and an axial resolution of about 1.6 microns. The Raman spectra were recorded using a Solar TII monochromator (SOL instruments GmbH, Augsburg, Germany) and an Andor CCD array (Oxford Instruments, Belfast, UK).

## 3. Results

### 3.1. Structure of Elastomers Modified with Carbon Nanotubes and Micro-Sized Cu and Ni Particles

The SEM images of the synthesized MWCTS are shown in Figure 2a,b and the TEM in Figure 2c,d. The morphology of the MWCNTs synthesized on Co-Mo/Al_2_O_3_-MgO catalysts is shown in. Analysis of the structure of the obtained MWCNTs (Figure 2c,d) reveals that these are MWCNTs with a diameter of ~40 nm, containing nickel metal particles at their ends. The MWCNTs with an extended filamentous formation with a diameter of 15–20 nm, length of more than 2 µm, and self-represented morphology are shown in Figure 2a,b. The observed TEM details of the structure are shown in Figure 2c,d. Figure 2d shows the encapsulation of the catalyst in the inner cavity of a carbon nanotube. The analysis of Figure 2c,d allows us to conclude that the MWCNTs are carbon nanotubes with a diameter of 10–15 nm. The metal particles of the catalyst (coal) are in the encapsulated state (inside the nanotubes). Figure 2e,f demonstrate the pattern of the MWCNTs’ distribution in the elastomer.

The distribution of the MWCNTs shown in Figure 2e,f has the character of a localized concentration with pronounced areas of unfilled polymer—with a level size of several micrometers, which is associated with the effect of the agglomeration of MWCNTs. In several reports [40,41], studies have been conducted related to the possibility of eliminating the agglomeration of MWCNTs in polymers. MWCNTs tend to form agglomerates and bundles in both liquid and solid media due to the high electron delocalization caused by the van der Waals interactions between the nanotubes [45,46].

Figure 3 demonstrates the MWCNTs’ size distribution. According to the analysis using the dynamic light scattering method, the MWCNTs in the range up to 22 μm are occupied by particles with a size of 6.2 μm, and particles with a size of 2.5 μm provide the greatest intensity. Moreover, 99% of the scattered radiation intensity is detected by particles up to 16 μm in size, while 90% of the total detectable intensity is accounted for by particles up to 5.7 μm in size.

It should be noted that the introduction of additional fillers can improve the interfacial strength of the polymer, reduce the interfacial tension, and improve the morphology by suppressing the tendency toward coalescence of the dispersed phase, which makes it possible to obtain highly efficient and multifunctional composites that can be used in a wide range of applications [47].

Figure 4a,b exhibit the distribution of the micro-sized nickel particles in the structure of a modified elastomer. Following the data in Figure 4a (SEM), two types of sizes up to 5 μm and up to 10–15 μm (Figure 4e) are characteristic of micro-sized Ni. Figure 4c,d show the distribution of micro-sized copper in the structure of a modified elastomer. According to Figure 2d, for Cu, two types of sizes of around 6–8 μm and around 15–20 μm are the most common (Figure 4f).

The analysis of the elemental composition showed that the elastomer with MWCNTs and Ni had Al particles in addition to Ni (Table 2 and Figure 3a). The presence of Si corresponded to the chemical composition of the elastomer, which had polar Si–O bonds. At the same time, carbon (C–C) was identified with the MWCNTs.

Table 3 shows the elemental composition of a modified elastomer with Cu and MWCNTs. The analysis of the elemental composition showed that the elastomer with MWCNTs and Si had in its composition, in addition to Cu and Al particles (Figure 5c and Table 3). The presence of Si corresponded to the chemical composition of the elastomer, which had polar Si–O bonds. At the same time, carbon (C–C) was identified with the MWCNTs.

The EDX of an elastomer with Ni and MWCNTs is shown in Figure 5a. Figure 5b shows the Raman spectrum of micro-sized Ni in the structure of a modified elastomer. The EDX of an elastomer with Cu and MWCNTs is shown in Figure 5c. According to Figure 5c, the structure of the material is dominated by Si, which belongs to the matrix of the organosilicon elastomer, and Cu is also present. Figure 5d shows the Raman spectrum of a modified elastomer with micro-sized Cu. It follows from Figure 5a that Si predominates in the structure of the modified composite, which belongs to the matrix of the organosilicon elastomer, and Ni also exists.

The Raman spectra (Figure 5b) are characterized by micro-sized Ni, MWCNTs, and elastomer matrix material. The Raman spectra (Figure 5d) are characterized by the formation of peaks characteristic of micro-sized Cu, MWCNTs and elastomer matrix. The Raman spectra have minimal differences, since the same carbon nanotubes are used to modify the elastomers. The typical D range, G range and 2D range, which are the main characteristics, are identical. The ratio of the intensities of these bands, D and G, corresponds to a value of 1.14, which is typical for multilayer carbon nanotubes and indicates that the structural properties of the carbon nanotubes did not significantly change during the formation of the two types of composites.

Figure 6a,b show 3D maps of the Raman spectra of the metal-filled sample surface and MWCNTs. Their analysis confirms that dispersed metal particles improve the uniformity of the MWCNTs’ distribution in the elastomer structure.

### 3.2. Electro and Thermophysical Studies of Elastomers with MWCNTs and Micro-Sized Ni and Cu Particles

Table 4 shows a comparative analysis of the effects of different concentrations of MWCNTs and micro-dimensional structures of Cu and Ni. 

It follows from the analysis of the data in Table 4 that the optimal combination is the mass concentration of MWCNT of 3% and Cu and Ni of 5%. With other combinations, i.e., with an increase in the concentration of MWCNT by 1% and Cu and Ni by 5%, there are no significant changes in the electrical conductivity. At the same time, an increase in the concentration of the conductive structures will lead to a deterioration in the physical and mechanical characteristics and, in particular, the flexibility and elasticity, and it will also not allow operation at a voltage of 220 V, since the supply current will be increased.

The assessment of the resistance to electrical voltage (220 V) was carried out through a comparison of elastomers with and without additives of micro-sized metals. Figure 7a–c show the temperature dependence of the power in the range from −40 to 40 °C with the repetition of on/off cycles of 50 and 100 cycles.

According to Figure 7a, the repeated switching on/off for 50 and 100 cycles leads to the degradation of the properties of the conductive elastomer from 1.1 kW/m^2^ (−40 °C) and, as a consequence, the power reduction to 0.3 kW/m^2^ (−40 °C) after 100 on/off cycles. At the same time, the addition of Ni (Figure 7b) allows an increase in power of 1.4 kW/m^2^ (−40 °C) and reduces the intensity of the degradation of the conductive structures (after 100 on–off cycles to 1.2 kW/m^2^ (−40 °C)). When changing Ni to Cu, the power of the modified composite in the heating mode increases to 1.6 kW/m^2^ (−40 °C). At the same time, the degradation of the conductive structures in the composite decreases in the mode of cyclic inclusions of 50 and 100 cycles (1.5 kW/m^2^ (−40 °C)). It should be noted that all the samples are characterized by the presence of the effect of temperature self-regulation, which follows from the decrease in the power of heat release when the temperature rises and the presence of a temperature (40 °C) at which the elastomer-based heater is completely turned off, which manifests in a decrease in the consumption of electrical energy. The studies of the temperature field distribution of the elastomer samples at a voltage of 220 V are exhibited in Figure 8.

The best result in terms of heat dissipation is typical for an elastomer sample with MWCNTs and copper (the temperature reaches 90.7 °C), which indicates the intensification of heat removal from the most overheated places in the composite structure. At the same time, the maximum temperature for Ni reaches 82.4 °C. The local unevenness of the temperature field distribution is characteristic of the sample without the addition of a micro-sized metal, which causes negative internal heating and destruction of the conductive structures based on the MWCNTs. The maximum temperature in this case reaches a value of 47.2 °C. Micro-dimensional structures allow reduction of the percolation threshold in terms of the electrical conductivity, which improves the overall picture of the temperature field distribution. It should be noted that the formation of an improved heat sink from locations with MWCNTs in an elastomer increases their resistance to electrophysical degradation and, accordingly, the heat output power is stabilized after multiple on and off cycles.

The addition of MWCNTs to the organosilicon matrix leads to a change in the molecular structure, which is associated with a change in the intensity of the peaks on the IR spectrogram. The IR spectra of the composite are shown in Figure 9. However, in the electric heating mode, the peaks of the IR spectrum do not change for any compositions with micro-sized metals.

## 4. Discussion

From a comparison of the modified composites based on dispersed micro-dimensional metals (Ni and Cu) and MWCNTs with existing analogs [43,44,45,46,47,48] (Table 5), it follows that the best electrophysical modes are associated with the ability to work for heating at a voltage of 220 V with a heat release temperature of 90.7 °C, which is higher than that of similar heaters [43,44,45,46]. At the same time, heat-generating heaters up to 160 and 310 °C at a voltage of 2.5 and 60 V consume a large current, which may result from the increased degradation of the heater material. At the same time, the transition to an elastomer matrix with Ni additives will allow the heater to be used at a lower temperature of 82.4 °C, which can be used in the selection of optimal modes for thermal ventilation or drying equipment. At the same time, the voltage at the level of 220 V allows you to reduce the current in comparison with the power supply at lower voltages (from 60 to 2.5 V) while maintaining a comparable or greater heat output power.

It should be noted that the degradation of the polymer matrix can be prevented by changes in the electrophysical parameters of the matrix itself by introducing various types of conductive structures adapted to the operating parameters [54], which is fully consistent with the concept of the conducted studies of the effect of micro-dimensional conductive structures on the stabilization of the electrophysical parameters of modified elastomers. In addition to improving the parameters of electrophysical stability, microparticles can provide thermal stabilization of the polymer matrix [55], which thereby significantly improves the properties of the modified elastomers used in the practice of obtaining heaters for various technical tasks.

## 5. Conclusions

The obtained data allow us to conclude the following:Samples of a heater modified with MWCNTs, synthesized with a Co-Mo/Al_2_O_3_-MgO catalyst, as well as using two types of micro-dimensional additives based on nickel (Ni) and copper (Cu) and an organosilicon compound, were obtained. The analysis (EDX) of the elemental composition showed that the elastomer (Ni) with MWCNTs and Ni had Al particles in addition to Ni. The analysis (EDX) of the elemental composition showed that the elastomer (Cu) with MWCNTs and Si had in its composition, in addition to Cu and Al particles. The presence of Si corresponded to the chemical composition of the elastomer, which had polar Si–O bonds. At the same time, carbon (C–C) was identified with the MWCNTsIt was found that the repeated switching on/off of the elastomer with MWCNTs at 50 and 100 cycles led to the degradation of the properties of the conductive elastomer from the available power of 1.1 kW/m^2^ (−40 °C) and, as a consequence, a power reduction to 0.3 kW/m^2^ (−40 °C) after 100 on/off cycles. At the same time, the addition of Ni (Figure 7b) allowed us to increase the power to 1.4 kW/m^2^ (−40 °C) and reduced the intensity of the degradation of the conductive structures (after 100 on–off cycles to 1.2 kW/m^2^ (−40 °C)). When changing Ni to Cu, the power of the modified composite in the heating mode increases to 1.6 kW/m^2^ (−40 °C), and at the same time, the degradation of the conductive structures in the composite is reduced in the mode of cyclic inclusions of 50 and 100 cycles (1.5 kW/m^2^ (−40 °C)).It was found that the best result in terms of heat dissipation has been obtained in the case of the elastomer containing MWCNTs and Cu (the temperature reaches 90.7 °C), which indicates the intensification of heat removal from the most overheated places in the composite structure. At the same time, the maximum temperature for Ni reaches 82.4 °C. The local non-uniformity of the temperature field distribution is characteristic of a sample without micro-sized metal additives, which causes negative internal overheating and destruction of the conductive structures based on the MWCNTs. The maximum temperature in this case reaches a value of 47.2 °C.

The relationship between the structural properties of the materials and their electrophysical characteristics was essential for the change in the thermal power of the composites, which follows from the analysis of SEM and EDX samples of the composites. The dispersed Cu included larger particles that were at least two times larger than the Ni particles. At the same time, the thermal conductivity of Cu is 389.6 W/m×°C and Ni is 90.9 W/ m×°C. The larger Cu particles had a better distribution with full coverage of the entire volume of the composite. For micro-sized Ni, two types of sizes are characteristic—up to 5 μm and up to 10–15 μm, and for Cu, two types of sizes are most common: about 6–8 μm and about 15–20 μm.

We anticipate reporting progress in developing technologies based on the reported observations.

## Figures and Tables

**Figure 1 polymers-16-00774-f001:**
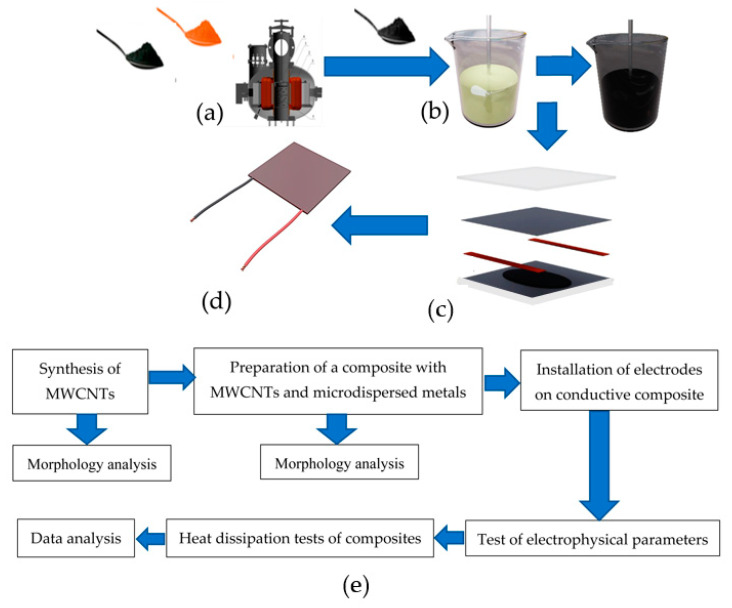
A scheme for the molding of a heater sample. (**a**)—mechanical mixing of micro-sized metal and MWCNTs; (**b**)—mechanical mixing of elastomer components A and B with a mixture of micro-sized metal and MWCNTs; (**c**)—molding of the composite with electrodes; (**d**)—general view of the heating element with wires: (**e**)—scheme for obtaining the composites and analyzing their properties.

**Figure 2 polymers-16-00774-f002:**
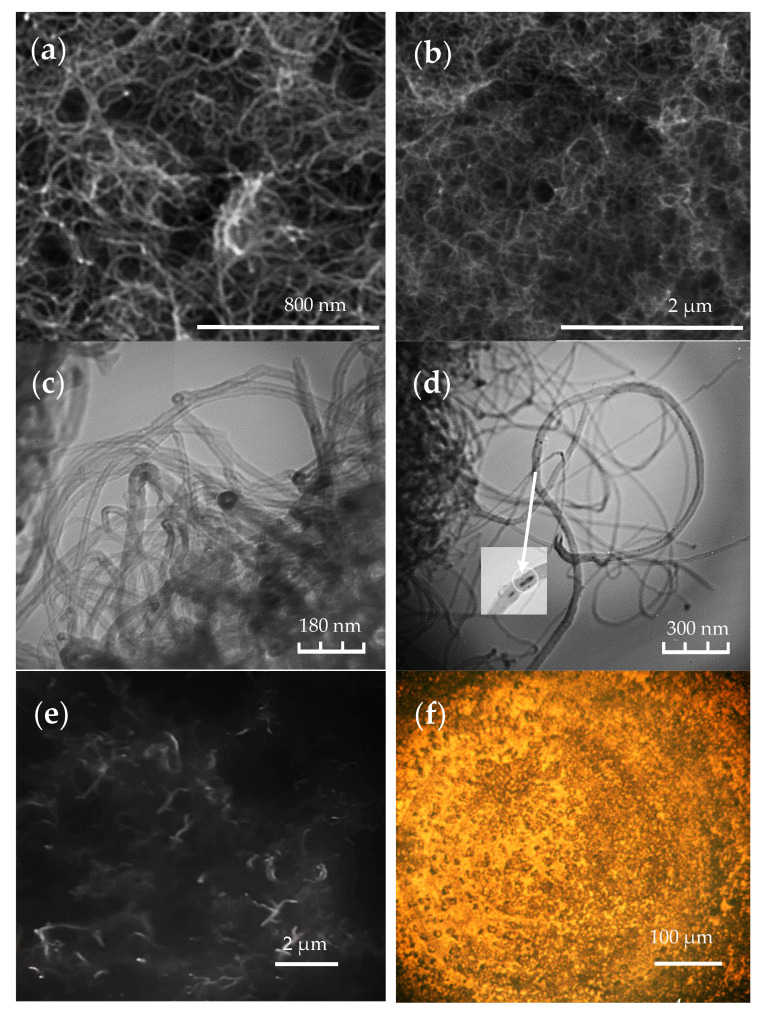
Microscopy data: (**a**,**b**)—SEM of MWCNTs synthesized on a Co-Mo/Al_2_O_3_-MgO catalyst; (**c**,**d**)—TEM of MWCNTs synthesized on a Co-Mo/Al_2_O_3_-MgO catalyst; (**e**)—SEM of MWCNTs’ distribution in the elastomer; (**f**)—optical image of MWCNTs’ distribution in the elastomer.

**Figure 3 polymers-16-00774-f003:**
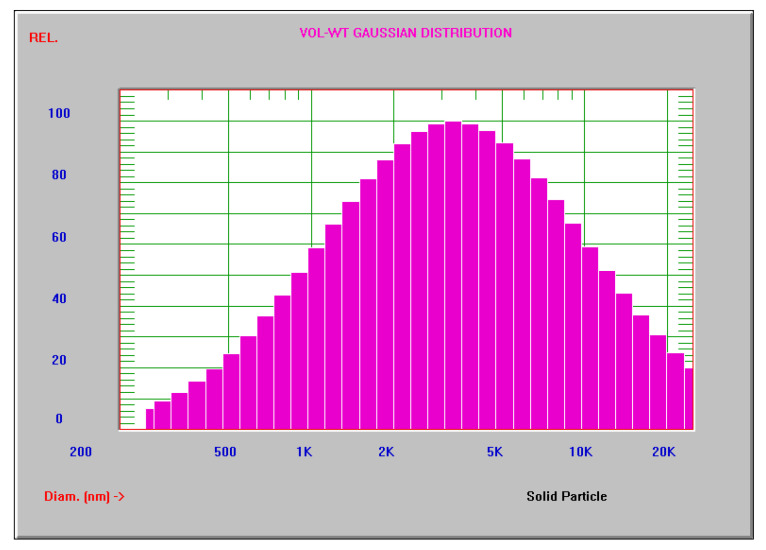
MWCNTs’ size distribution.

**Figure 4 polymers-16-00774-f004:**
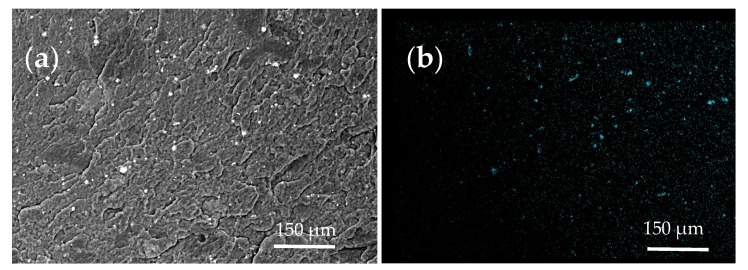
Microscopy of elastomer(s) with Ni and MWCNTs: (**a**)—SEM snapshot; (**b**)—EDX distribution map of Ni; with Cu and MWCNTs: (**c**)—SEM snapshot; (**d**)—EDX distribution map of Cu; (**e**)—distribution map of composite with Ni and MWCNTs; (**f**)—distribution map of composite with Cu and MWCNTs.

**Figure 5 polymers-16-00774-f005:**
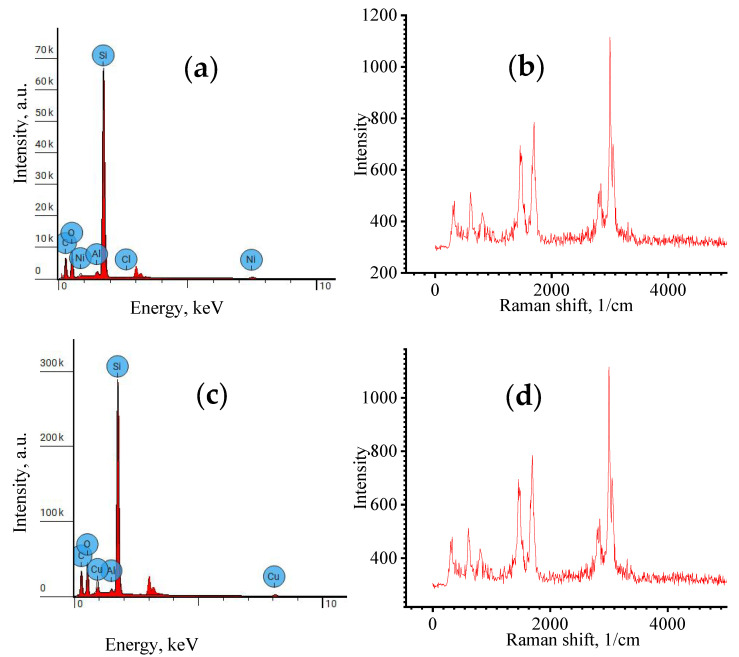
(**a**)—EDX of elastomer with Ni and MWCNTs; (**b**)—Raman spectra of elastomer with MWCNTs and Ni; (**c**)—EDX of elastomer with Cu and MWCNTs; (**d**)—Raman spectra of elastomer with Cu and MWCNTs. For a detailed explanation of proximity panels (**a**–**c**) and (**b**–**d**), see below.

**Figure 6 polymers-16-00774-f006:**
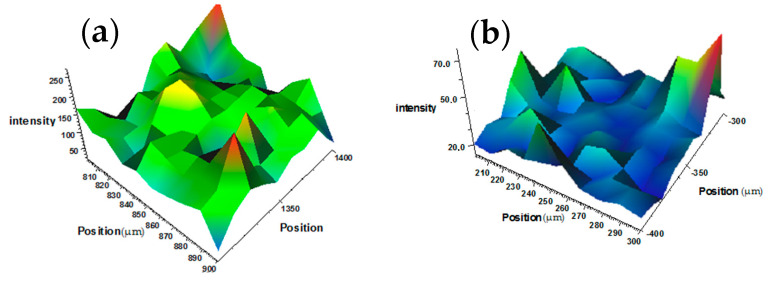
3D maps of Raman spectra: (**a**)—sample surfaces with metal filling and MWCNTs (green - polymer matrix; red and blue - inclusions of conductive structures); (**b**)—sample surfaces with MWCNT filling (blue - polymer matrix; red and green - inclusions of conductive structures).

**Figure 7 polymers-16-00774-f007:**
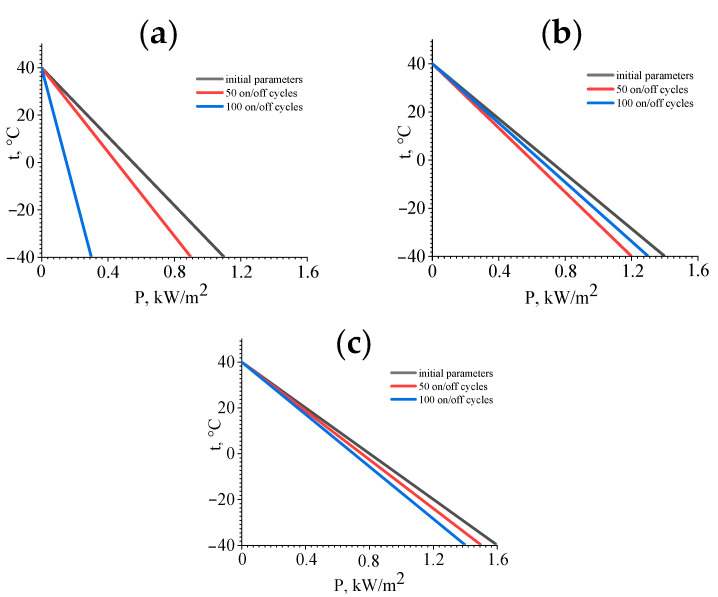
Specific power of heat: (**a**)—composite with MWCNTs only; (**b**)—composite with Ni and MWCNTs; and (**c**)—composite with Cu and MWCNTs.

**Figure 8 polymers-16-00774-f008:**
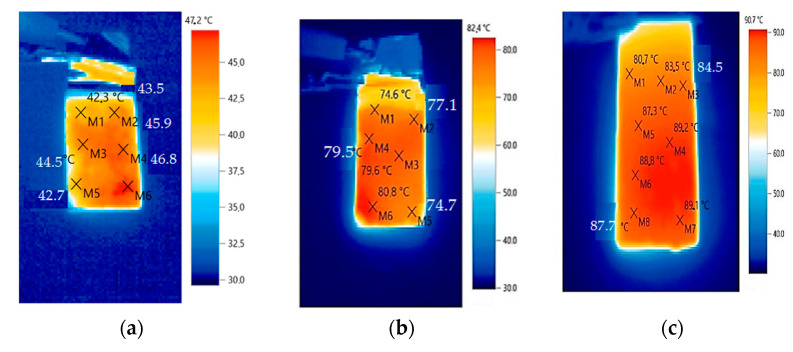
Thermograms of samples: (**a**)—elastomer with MWCNTs; (**b**)—elastomer with MWCNTs and Ni; (**c**)—elastomer with MWCNTs and Cu.

**Figure 9 polymers-16-00774-f009:**
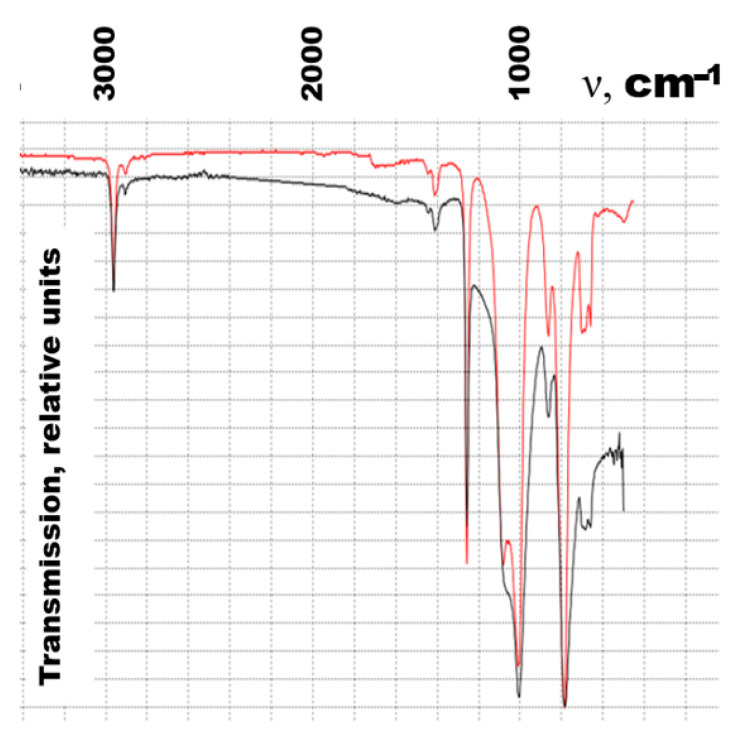
IR spectra of the composite with (black) and without MWCNTs (red).

**Table 1 polymers-16-00774-t001:** Characteristics of the Co-Mo/Al_2_O_3_-MgO catalyst.

Specific Surface Area, m^2^/g	Efficiency, g_MWCNTs_/g_catalyst_
82.3	21.15

**Table 2 polymers-16-00774-t002:** The composition of an elastomer with Ni and MWCNTs.

Element	Atomic %	Weight %
C	52.700	36.236
O	20.107	18.418
Al	0.713	1.101
Si	25.458	40.941
Cl	0.099	0.200
Ni	0.924	3.103

**Table 3 polymers-16-00774-t003:** Elemental composition of an elastomer with copper and MWCNTs.

Element	Atomic %	Weight %
C	53.920	36.963
O	20.346	18.581
Al	0.584	0.899
Si	23.552	37.762
Cu	1.598	5.794

**Table 4 polymers-16-00774-t004:** Specific volume and surface resistance.

№	Composition	Specific Volume Resistance δ_M_, Om × m	Specific Surface Resistance R_yд.HOB_, кOm
1	MWCNT 2%, Ni 5%	85.8	250
2	MWCNT 3%, Ni 5%	69.15	240
3	MWCNT 4%, Ni 5%	69	230
4	MWCNT 3%, Ni 10%	67.75	210
5	MWCNT 2%, Cu 5%	57.2	330
6	MWCNT 3%, Cu 5%	34.9	320
7	MWCNT 4%, Cu 5%	33.2	310
8	MWCNT 3%, Cu 10%	31.54	300

**Table 5 polymers-16-00774-t005:** Comparison of the obtained electrothermal properties (*) with the reports on different CNT-containing materials.

№	Material	Voltage (V)	Temperature (°C)
1	CNT [48]	2.5	70.4
2	CNT [49]	5	50
3	Carbon fiber (CF)/asphalt mastics [50]	60	5
4	CNT [51]	35	75
5	CNT Film [52]	2.5	310
6	SWCNT [53]	60	160
7	(*)	220	93.9 (Cu)
		220	86.7 (Ni)

## Data Availability

The data presented in this study are available on reasonable request.

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
