# Peer review of "Properties of Organosilicon Elastomers Modified with Multilayer Carbon Nanotubes and Metallic (Cu or Ni) Microparticles"

_polymers, 2024, doi:10.3390/polym16060774_

Round 1

Reviewer 1 Report

Comments and Suggestions for Authors

The authors presented a study about the electro-thermophysical characteristics of organosilicon elastomers modified with MWCNT and metallic (Cu or Ni) microparticles. This is an interesting topic. However, authors investigated only 3 cases (3wt%MWCNT / elastomer, 3wt%MWCNT / 5wt%Ni / elastomers, 3wt%MWCNT / 5wt%Cu / elastomers composites). Then, they reported very few results as following: “The heat dissipation has been obtained in the case of elastomer containing MWCNTs and Cu (the temperature reaches 90.7 ° C), which indicates the intensification of heat removal from the most overheated places in the composite structure. At the same time, the maximum temperature for Ni reaches 82.4 ° C. The local non-uniformity of the temperature field distribution is characteristic of a sample without micro-sized metal additives, which forms a negative internal overheating and destruction of conductive structures based on MWCNTs. The maximum temperature in this case reaches a value of 47.2 ° C.” They showed the improvement without a detailed explanation. I would like to suggest that the authors have to investigate more study cases to prove the clear effect of metallic microparticles on electro-thermophysical properties. And the relationship between electro-thermophysical properties and structural properties should be discussed in more detail.

I also would like to suggest that the authors should consider the other aspects to improve the quality of the manuscript.

1.       In the Materials and Methods section, authors should briefly describe the sample preparation method, even though it is almost the same as their previous studies. And the main stages of obtaining a nanomodified composite should be rewritten to be easier to understand.   

2.       Why did the authors select 3 wt% of MWCNT and 5 wt% of metallic particles to prepare composite samples? As I mentioned above, the authors should add more study cases to prove the behavior and why it happened.

3.       The SEM images in Fig. 1 were of low magnitude and low resolution. The results of Fig. 1e and f do not demonstrate the pattern of MWCNTs distribution on elastomers.

4.       In Fig. 2, the results showed the metallic particles seemed to agglomerate in elastomers. Please describe the particle size before mixing and after mixing and metallic distribution on elastomers.

5.       Authors analyzed the composition of composites using EDX. It showed that Ni weight content is about 3.1 wt%, whereas Cu weight content is about 5.8 wt%. The authors cannot compare their electro-thermophysical properties as their content are not similar. Thus, the conclusion of this manuscript cannot convince the audience.

6.       There is no difference between the Raman spectra of 3wt%MWCNT / 5wt%Ni / elastomers, 3wt%MWCNT / 5wt%Cu / elastomers composites. Why did the author use this result in the manuscript?

Comments on the Quality of English Language

Some grammar errors and mistyping words were found. Please check carefully the manuscript again.

Author Response

2024-02-06

To Reviewer1

Dear Reviewer:

Thank you so much for your efforts to improve our contribution. As about your comments, let me respond on them according to their order in your review:

  1. In the Materials and Methods section, authors should briefly describe the sample preparation method, even though it is almost the same as their previous studies. And the main stages of obtaining a nanomodified composite should be rewritten to be easier to understand.   

Corrected

  1. Why did the authors select 3 wt% of MWCNT and 5 wt% of metallic particles to prepare composite samples? As I mentioned above, the authors should add more study cases to prove the behavior and why it happened.

From the data in Table 4 one can see, that the optimal combination is the mass concentration of MWCNT – 3% and copper and nickel 5%. With other combinations, i.e., with an increase in the concentration of MWCNT by 1% and copper or nickel by 5%, there are no significant changes in electrical conductivity. At the same time, an increase in the concentration of conductive structures will lead to a deterioration in physical and mechanical characteristics, and in particular flexibility and elasticity, and will also not allow operation at a voltage of 220 V, since the supply current will be increased.

  1. The SEM images in Fig. 1 were of low magnitude and low resolution. The results of Fig. 1e and f do not demonstrate the pattern of MWCNTs distribution on elastomers.

Corrected accordingly

  1. In Fig. 2, the results showed the metallic particles seemed to agglomerate in elastomers. Please describe the particle size before mixing and after mixing and metallic distribution on elastomers.

Dispersed copper consists of larger particles that were at least 2 times larger than nickel particles. The larger copper particles had a better distribution with full coverage of the entire volume of the composite. For micro-sized nickel, size distribution looks bimodal with peaks around up to 5 microns and up to 10-15 microns, whereas for copper peaks are around about 6-8 microns and about 15-20 microns.

Dispersion does not significantly affect the size and agglomeration of metal particles. Metals do not have van der Waals forces. It is enough to carry out mechanical mixing in a liquid polymer during the time specified in the procedure. At the same time, metal particles affect the reduction of van der Waals forces for CNTs, which reduces agglomeration.

  1. Authors analyzed the composition of composites using EDX. It showed that Ni weight content is about 3.1 wt%, whereas Cu weight content is about 5.8 wt%. The authors cannot compare their electro-thermophysical properties as their content are not similar. Thus, the conclusion of this manuscript cannot convince the audience.

EDX analysis covers a local area. We anticipate that over the volume will be equality. The mass concentration is the same, which allows for an adequate assessment.

  1. There is no difference between the Raman spectra of 3wt%MWCNT / 5wt%Ni / elastomers, 3wt%MWCNT / 5wt%Cu / elastomers composites. Why did the author use this result in the manuscript?

This was done in order to show that conductive additives have the same effect on the elastomer matrix. And if degradation of the composite occurs, then the cause is the destruction of CNTs.

Once again thank you for your impact and please let us know if we could do something else to improve the quality of current submission.

Sincerely

Dr. Alex Vetcher

Reviewer 2 Report

Comments and Suggestions for Authors

Meritoric advice:

1. " (...) and corrosion resistance [2,3] and can be used 46 in conditions of increased vibration" is not applied for pluriver polymers but for selected ones, and this should be underlined in the text!

2. "(...) a good ability to absorb an 50 electromagnetic field" there is more appropriate word to use here: EM shielding.

3.  There are mentioned "micro-sized metals 95 Ni and Cu" and " Ni and Cu micro-particles." but there is no information what are the differences between them. It is highly needed here! If there are the same it is inconsistent by author to mix-up the definitions!

4. "The use of the obtained catalyst in the CVD process (carbon source propane-butane 104 mixture, t=650 °C, 40 min) makes it possible to obtain a homogeneous nano-structural 105 material with a specific surface area of 306.2 m2 /g. These catalysts were used for the syn- 106 thesis of MWCNTs by the CVD method." The gas is a source of carbon to grow CNTs, a catalyst must be sth else. From the text you mentioned catalyst later but in your ay of description it seems the catalyst is a gas at the first statements not a metallic film that appeared afterward. Please make it straight! Otherwise for me as the reviewer it is clear the authors do not understand the synthesis method. Knowing it is crucial for your research on composite material!

5. " used the ultra- 118 sonic dispergator" - the authos shoudl clearly state if they use US for CNTs or metallic particles/powder or in combination with elastomer, and what parameters they set.

6. an accelerating voltage up to 200 keV is really destroyable for CNTs. The authors. 

7. It is not visible:  "Metal particles of the catalyst (coal) are in the encapsulated state (inside nanotubes)". Please make it shown clearly.

8. Fig. 1 e, f - the matrix structure not clearly visible. Make it please visible. Work on image correction maybe by use of contrast e.g. or take please another TEM/SEM images.

9. There are missing data with the images of metallic particles itselves and size distribution graphs to be reference to the composite matrix. 

10. "Figure 2.a,b exhibits the distribution of micro-sized nickel particles" - the description says about Cu only!

11. The authors discuss Ni and Cu particles but Ni is missing in Table and actually in EDX as well, why?

12. It is not clear what for Fig. 4 is released and what information it really gives to the readers.

13. "on dispersed micro-di- 276 mensional metals (Ni and Si) " - ??? not NI and Cu?

Generally, the authors should focus on what is the purpose of the research here and stay it clearly at the introduction!

The resistance and IR spectroscopy measurement would help here to explain also the matter of issues related to the thermal degradation of the composite.

Comments on the Quality of English Language

Some grammatical error need to be corrected, e.g. the reviewer found:

1. " (-40°CC) " or " of the sample samples and on several samples -" check the proper writing across the whole document please - more such stuff present!

2.  "Carbon nano-tubes (CNTs) [14]" -> carbon nanotubes OK!

Generally English understood but should be improved and checked.

Author Response

2024-02-06

To Reviewer2

Dear Reviewer:

Thank you so much for your efforts to improve our contribution. As about your comments, let me respond on them according to their order in your review:

  1. " (...) and corrosion resistance [2,3] and can be used 46 in conditions of increased vibration" is not applied for pluriver polymers but for selected ones, and this should be underlined in the text!

Corrected

  1. "(...) a good ability to absorb an 50 electromagnetic field" there is more appropriate word to use here: EM shielding.

Corrected

  1. There are mentioned "micro-sized metals 95 Ni and Cu" and " Ni and Cu micro-particles." but there is no information what are the differences between them. It is highly needed here! If there are the same it is inconsistent by author to mix-up the definitions!

Corrected accordingly

  1. "The use of the obtained catalyst in the CVD process (carbon source propane-butane 104 mixture, t=650 °C, 40 min) makes it possible to obtain a homogeneous nano-structural 105 material with a specific surface area of 306.2 m2 /g. These catalysts were used for the syn- 106 thesis of MWCNTs by the CVD method." The gas is a source of carbon to grow CNTs, a catalyst must be sth else. From the text you mentioned catalyst later but in your ay of description it seems the catalyst is a gas at the first statements not a metallic film that appeared afterward. Please make it straight! Otherwise for me as the reviewer it is clear the authors do not understand the synthesis method. Knowing it is crucial for your research on composite material!

It was clarified

  1. " used the ultra- 118 sonic dispergator" - the authos shoudl clearly state if they use US for CNTs or metallic particles/powder or in combination with elastomer, and what parameters they set.

Corrected

  1. an accelerating voltage up to 200 keV is really destroyable for CNTs. The authors. 

Maybe it was due to the right heat dissipation

  1. It is not visible:  "Metal particles of the catalyst (coal) are in the encapsulated state (inside nanotubes)". Please make it shown clearly.

Corrected

  1. 1 e, f - the matrix structure not clearly visible. Make it please visible. Work on image correction maybe by use of contrast e.g. or take please another TEM/SEM images.

Corrected

  1. There are missing data with the images of metallic particles itselves and size distribution graphs to be reference to the composite matrix. 

In the dispersed state, metal particles experience compactization. They are difficult to visualize separately, therefore, the distribution is also poorly displayed. But in the polymer itself, you can see their size and ability to be distributed in a given concentration.

  1. "Figure 2.a,b exhibits the distribution of micro-sized nickel particles" - the description says about Cu only!

Corrected

  1. The authors discuss Ni and Cu particles but Ni is missing in Table and actually in EDX as well, why?

Corrected

  1. It is not clear what for Fig. 4 is released and what information it really gives to the readers.

To visually evaluate the free space

  1. "on dispersed micro-di- 276 mensional metals (Ni and Si) " - ??? not NI and Cu?

Corrected

  1. Generally, the authors should focus on what is the purpose of the research here and stay it clearly at the introduction! The resistance and IR spectroscopy measurement would help here to explain also the matter of issues related to the thermal degradation of the composite.

Corrected

  1. Some grammatical error need to be corrected, e.g. the reviewer found:
  2. " (-40°CC) " or " of the sample samples and on several samples -" check the proper writing across the whole document please - more such stuff present!
  3. "Carbon nano-tubes (CNTs) [14]" -> carbon nanotubes OK!

Generally English understood but should be improved and checked.

We edited the body and invited Native American speaker to improve the readability.

Once again thank you for your impact and please let us know if we could do something else to improve the quality of current submission.

Sincerely

Dr. Alex Vetcher

Reviewer 3 Report

Comments and Suggestions for Authors

The article titled "Properties of organosilicon elastomers modified with multi-layer carbon nanotubes and metallic (Cu or Ni) microparticles" claims to assess the properties of organosilicon elastomers with carbon nanotubes and metallic particles and their ability to control degradation under electrical voltage. Accordingly, I assess that the objective of the work is within the scope of the journal. However, based on the following observations, I respectfully recommend against publishing the work in its current version.

 - The results presented in section 3.2 evaluate the temperature dependence of power and the heat dissipation capacity of the samples. However, there was no analysis of the electrical/thermal conductivity of the samples with and without the addition of metallic particles. From the SEM micrographs, although inconclusive, it is possible to observe poor dispersion of MWCNTs, forming large clusters. This negatively impacts the formation of an interconnected network of MWCNTs within the polymer. Thus, the analysis of heat dissipation may be conducted in an insulating system (elastomer with MWCNT below the percolation threshold) and a conductive system (with the addition of metallic particles), rendering the comparison parameters ineffective.

- Another important point to be analyzed is the dispersion of particles in the polymeric matrix, which directly impacts conductivity. In particular, the impact of adding metallic particles on the dispersion of MWCNTs. As mentioned, the presented micrographs are inconclusive, especially those obtained by SEM of the nanocomposite.

- Throughout the work, there are several mistaken references to the term "nano" when referring to the composite that uses metallic particles on a micro-scale.

 - In the introduction, the relationship between the work and "electrically conductive polymeric matrices" mentioned in line 48 is unclear. The matrix used has insulating characteristics, correct? Subsequently, the broad range of functional properties of these matrices (line 50) needs clarification.

- In line 51, I believe the term "smart polymers" would be more appropriate. However, smart polymers are an extremely broad class, and when not described, it may sound like a generic term. In which category does the work belong?

- In line 60, there is the assertion that "carbon and metal fillers can be combined in a polymer composite." They can be classified as hybrid (nano)composites. However, these systems, to which the material of the work belongs, are not thoroughly discussed, indicating their state of the art.

- The introduction of a dispersed metal into the polymer is described in line 62, but the indicated reference (ref. 22) is a study on polymer metallization and not necessarily on the addition of (nano) metallic particles (in the form of a (nano) composite).

 - In the methodology, references are needed for the CVD process and the properties of materials from the process. Additionally, the data in Table 1, especially specific surface area, requires a detailed description of its acquisition.

- The obtaining of composites needs a more elaborate description. In line 117, details on the frequency, time, and temperature used in the ultrasonic bath for MWCNT dispersion are crucial. Since these are simple concentrations, using equations may be unnecessary. A more detailed procedure description, possibly with a simple table displaying concentrations for each sample, would suffice.

- In line 139, the methodology lacks information on how and where the power supply was used.

- The use of Raman spectroscopy is not described in the methodology.

- The description of TEM/SEM is confusing and contains errors. Furthermore, the SEM description for the elastomer does not specify how the sample was prepared for SEM observation.

 - In the results, the SEM/TEM micrographs are of low resolution/quality, hindering a successful material analysis. Moreover, the SEM/TEM discussion is confusing, reporting different diameters for the same set of images. A more detailed discussion is needed, along with an appropriate distribution curve if there is a diameter distribution, along with a description of the procedure.

- Figure 2 presents micrographs only at low magnification, possibly for EDX analysis. However, a morphological analysis of the systems is crucial. Additionally, a size distribution curve of Ni and Cu in the elastomer would be relevant.

- In line 229, the discussion of Raman results is inconclusive and lacks mention of peaks obtained in the spectrum. Additionally, observations do not refer to the literature. In line 233, the uniformity of MWCNT distribution based on Figure 4 is unclear and contradicts SEM observations.

 - In the discussion section, it is pertinent to review the discussion and data presented in Table 4. For instance, ref. 42 is a review article, making the reported data potentially inappropriate for this work. Meanwhile, ref. 43 utilizes a graphene/Fe-loaded polyester fabric-PET.

Author Response

2024-02-06

To Reviewer3

Dear Reviewer:

Thank you so much for your efforts to improve our contribution. As about your comments, let me respond on them according to their order in your review:

  1. The results presented in section 3.2 evaluate the temperature dependence of power and the heat dissipation capacity of the samples. However, there was no analysis of the electrical/thermal conductivity of the samples with and without the addition of metallic particles. From the SEM micrographs, although inconclusive, it is possible to observe poor dispersion of MWCNTs, forming large clusters. This negatively impacts the formation of an interconnected network of MWCNTs within the polymer. Thus, the analysis of heat dissipation may be conducted in an insulating system (elastomer with MWCNT below the percolation threshold) and a conductive system (with the addition of metallic particles), rendering the comparison parameters ineffective."(...)

The heat dissipation in MWCNT-containing composites had been reported in e.g.:

Ali, I.; Shchegolkov, A.; Shchegolkov, A.;  Zemtsova, N.; Bogoslovskiy, V.; Shigabaeva, G.; Galunin, E.; Hussain, I.; Almalki, A.S.A.; Alsharif, M.A.; Alahmdi, M.I. Preparation and application practice of temperature self‐regulating flexible polymer electric heaters. Polym.Eng. Sci. 2022, 62(3), 730. [CrossRef]

Shchegolkov, A.V.; Jang, S.-H.; Shchegolkov, A.V.; Rodionov, Y.V.; Glivenkova, O.A. Multistage Mechanical Activation of Multilayer Carbon Nanotubes in Creation of Electric Heaters with Self-Regulating Temperature. Materials. 2021, 14, 4654. [CrossRef]

This study makes it possible to improve the behavior of composites by forming an improved heat sink. Dispersed copper included larger particles that were at least 2 times larger than nickel particles. At the same time, the thermal conductivity of copper is 389.6 W/m°C, and nickel is 90.9 W/m°C.

In Figure 2, a composite SEM has been added, where there is a view with a general distribution of MWCNT. At the same time, metal particles affect the reduction of van der Waals forces for CNTs, which reduces agglomeration.

  1. Another important point to be analyzed is the dispersion of particles in the polymeric matrix, which directly impacts conductivity. In particular, the impact of adding metallic particles on the dispersion of MWCNTs. As mentioned, the presented micrographs are inconclusive, especially those obtained by SEM of the nanocomposite.

We modified Figure 2 accordingly

  1. Throughout the work, there are several mistaken references to the term "nano" when referring to the composite that uses metallic particles on a micro-scale.

Corrected

  1. In the introduction, the relationship between the work and "electrically conductive polymeric matrices" mentioned in line 48 is unclear. The matrix used has insulating characteristics, correct? Subsequently, the broad range of functional properties of these matrices (line 50) needs clarification.

Corrected

  1. In line 51, I believe the term "smart polymers" would be more appropriate. However, smart polymers are an extremely broad class, and when not described, it may sound like a generic term. In which category does the work belong?

Corrected

  1. In line 60, there is the assertion that "carbon and metal fillers can be combined in a polymer composite." They can be classified as hybrid (nano)composites. However, these systems, to which the material of the work belongs, are not thoroughly discussed, indicating their state of the art.

Corrected

  1. The introduction of a dispersed metal into the polymer is described in line 62, but the indicated reference (ref. 22) is a study on polymer metallization and not necessarily on the addition of (nano) metallic particles (in the form of a (nano) composite).

Corrected

  1. In the methodology, references are needed for the CVD process and the properties of materials from the process. Additionally, the data in Table 1, especially specific surface area, requires a detailed description of its acquisition.

The specific surface of the MNT was studied using the 5-point Brunauer-Emmett-Teller (BET) method.

The synthesis of MNT is described in detail in the work: Ali, I.; Shchegolkov, A.; Shchegolkov, A.;  Zemtsova, N.; Bogoslovskiy, V.; Shigabaeva, G.; Galunin, E.; Hussain, I.; Almalki, A.S.A.; Alsharif, M.A.; Alahmdi, M.I. Preparation and application practice of temperature self‐regulating flexible polymer electric heaters. Polym.Eng. Sci. 2022, 62(3), 730. [CrossRef]

  1. The obtaining of composites needs a more elaborate description. In line 117, details on the frequency, time, and temperature used in the ultrasonic bath for MWCNT dispersion are crucial. Since these are simple concentrations, using equations may be unnecessary. A more detailed procedure description, possibly with a simple table displaying concentrations for each sample, would suffice."on dispersed micro-di- 276 mensional metals (Ni and Si) " - ??? not NI and Cu?

Corrected accordingly

  1. In line 139, the methodology lacks information on how and where the power supply was used.

Corrected

  1. The use of Raman spectroscopy is not described in the methodology.

Corrected

  1. The description of TEM/SEM is confusing and contains errors. Furthermore, the SEM description for the elastomer does not specify how the sample was prepared for SEM observation.

Corrected

  1. Figure 2 presents micrographs only at low magnification, possibly for EDX analysis. However, a morphological analysis of the systems is crucial. Additionally, a size distribution curve of Ni and Cu in the elastomer would be relevant.

Corrected accordingly

  1. In line 229, the discussion of Raman results is inconclusive and lacks mention of peaks obtained in the spectrum. Additionally, observations do not refer to the literature. In line 233, the uniformity of MWCNT distribution based on Figure 4 is unclear and contradicts SEM observations.

We added SEM and Raman to evaluate influence of metallic particles on the parameters of the matrix

  1. - In the discussion section, it is pertinent to review the discussion and data presented in Table 4. For instance, ref. 42 is a review article, making the reported data potentially inappropriate for this work. Meanwhile, ref. 43 utilizes a graphene/Fe-loaded polyester fabric-PET

Corrected

Once again thank you for your impact and please let us know if we could do something else to improve the quality of current submission.

Sincerely

Dr. Alex Vetcher

Round 2

Reviewer 1 Report

Comments and Suggestions for Authors

Thank you very much for taking the effort to revise your manuscript. In its current form, this paper can be considered for publication. 

Comments on the Quality of English Language

A comment is written in Russian in Table 4. Kindly, please fix it. 

Author Response

Dear Reviewer:

Thank you so much for your assistance to improve the quality of our submission. As about corrections - the were done accordingly.

Please let us know if we do something else to improve the quality of our submission.

Sincerely

Dr. Alex Vetcher

Reviewer 3 Report

Comments and Suggestions for Authors

Dear authors, I appreciate your attention to the responses provided to the inquiries. Below, I present my considerations:

1. I appreciate the response and the indication of associated literature. In the referenced works, the conductivity of nanocomposites is a crucial aspect. I suggest revisiting some points in the introduction, incorporating the cited works and others representing the state of the art, to clarify not only the objective of the study but especially what recent works have achieved in the field.

2. In fact, my reference was the SEM micrographs of the composites. Was Figure 2 supposed to depict micrographs of MWCNTs until now? Furthermore, TEM/SEM micrographs exhibit low resolutions.

5. It was not corrected or justified. The relation of the term to the study was also not explained.

6. There was no modification here. On the contrary, it seems that the sentence where I had commented on the need for adjustment was apparently reproduced. In this case, it was expected that hybrid systems would be exemplified based on the state-of-the-art literature and their connection to the present study.

7. The reference remains the same. However, the reference indicates metallized polymer systems, while the text refers to polymer composites obtained with "dispersed metal into the polymer structure."

13. In reality, there was no morphological evaluation of the composites in the current Figure 4. Additionally, the DLS distribution curve does not represent the composite system accurately, as the synthesis process involves dispersion and potential "breaks" in the length of MWCNTs.

Author Response

2024-02-22

To reviewer

Dear reviewer:

Thank you so much for your hard work to improve our submission. As about your comments, let me respond on them according to their appearance in your review:

#

Comment

Response

1

I appreciate the response and the indication of associated literature. In the referenced works, the conductivity of nanocomposites is a crucial aspect. I suggest revisiting some points in the introduction, incorporating the cited works and others representing the state of the art, to clarify not only the objective of the study but especially what recent works have achieved in the field.

Corrected. We added                    22.                Cho, Y. M., Lee, S.-S., Park, C. R., Kim, T. A. & Park, M. Enhanced electrical conductivity of polymer microspheres by altering assembly sequence of two different shaped conductive fillers. Compos. Pt A-Appl. Sci. Manuf. 149, 106562 (2021). [CrossRef]

23.          Hu, Ning, et al. "Ultrasensitive strain sensors made from metal-coated carbon nanofiller/epoxy composites." Carbon 51 (2013): 202-212.

24.          Lee, Jaewook, et al. "Magnetically aligned iron oxide/gold nanoparticle-decorated carbon nanotube hybrid structure as a hu-midity sensor." ACS applied materials & interfaces 7.28 (2015): 15506-15513.

25.          Safina, L., Baimova, J. & Mulyukov, R. Nickel nanoparticles inside carbon nanostructures: atomistic simulation. Mech Adv Mater Mod Process 5, 2 (2019).

26.          Wang, Y., Lu, S., He, W. et al. Modeling and characterization of the electrical conductivity on metal nanoparticles/carbon nanotube/polymer composites. Sci Rep 12, 10448 (2022).

27.          Sadek, E.M., Ahmed, S.M., Mansour, N.A. et al. Synthesis, characterization and properties of nanocomposites based on poly(vinyl chloride)/carbon nanotubes–silver nanoparticles. Bull Mater Sci 46, 30 (2023)

2

In fact, my reference was the SEM micrographs of the composites. Was Figure 2 supposed to depict micrographs of MWCNTs until now? Furthermore, TEM/SEM micrographs exhibit low resolutions.

We added Figure 2.e, f to demonstrate the pattern of MWCNTs distribution in the elastomer. The resolution allows you to evaluate their morphology and geometric parameters.

5

It was not corrected or justified. The relation of the term to the study was also not explained.

Corrected accordingly.

6

There was no modification here. On the contrary, it seems that the sentence where I had commented on the need for adjustment was apparently reproduced. In this case, it was expected that hybrid systems would be exemplified based on the state-of-the-art literature and their connection to the present study.

Corrected accordingly.

7

The reference remains the same. However, the reference indicates metallized polymer systems, while the text refers to polymer composites obtained with "dispersed metal into the polymer structure."

Corrected accordingly.        Take a look on                      22.          Cho, Y. M., Lee, S.-S., Park, C. R., Kim, T. A. & Park, M. Enhanced electrical conductivity of polymer microspheres by altering assembly sequence of two different shaped conductive fillers. Compos. Pt A-Appl. Sci. Manuf. 149, 106562 (2021). [CrossRef]

13

In reality, there was no morphological evaluation of the composites in the current Figure 4. Additionally, the DLS distribution curve does not represent the composite system accurately, as the synthesis process involves dispersion and potential "breaks" in the length of MWCNTs.

We added histograms. Figure 4 (e, f) and

Figure 2(e, f) demonstrate the pattern of MWCNTs distribution in the elastomer.

Please let us know if we do something else to improve the quality of our submission.

Sincerely

Dr. Alex Vetcher